# Evaluation of the Relation between Ictal EEG Features and XAI Explanations

**DOI:** 10.3390/brainsci14040306

**Published:** 2024-03-25

**Authors:** Sergio E. Sánchez-Hernández, Sulema Torres-Ramos, Israel Román-Godínez, Ricardo A. Salido-Ruiz

**Affiliations:** Division of Cyber-Human Interaction Technologies, University of Guadalajara (UdG), Guadalajara 44100, Mexico; sergio.sanchez1153@alumnos.udg.mx (S.E.S.-H.); sulema.torres@academicos.udg.mx (S.T.-R.); israel.roman@academicos.udg.mx (I.R.-G.)

**Keywords:** EEG, epilepsy, machine learning, explainable artificial intelligence, correlation

## Abstract

Epilepsy is a neurological disease with one of the highest rates of incidence worldwide. Although EEG is a crucial tool for its diagnosis, the manual detection of epileptic seizures is time consuming. Automated methods are needed to streamline this process; although there are already several works that have achieved this, the process by which it is executed remains a black box that prevents understanding of the ways in which machine learning algorithms make their decisions. A state-of-the-art deep learning model for seizure detection and three EEG databases were chosen for this study. The developed models were trained and evaluated under different conditions (i.e., three distinct levels of overlap among the chosen EEG data windows). The classifiers with the best performance were selected, then Shapley Additive Explanations (SHAPs) and Local Interpretable Model-Agnostic Explanations (LIMEs) were employed to estimate the importance value of each EEG channel and the Spearman’s rank correlation coefficient was computed between the EEG features of epileptic signals and the importance values. The results show that the database and training conditions may affect a classifier’s performance. The most significant accuracy rates were 0.84, 0.73, and 0.64 for the CHB-MIT, Siena, and TUSZ EEG datasets, respectively. In addition, most EEG features displayed negligible or low correlation with the importance values. Finally, it was concluded that a correlation between the EEG features and the importance values (generated by SHAP and LIME) may have been absent even for the high-performance models.

## 1. Introduction

Epilepsy is a neurological disease that presents recurrent and unprovoked seizures [1]. This disease affects the physical health of the patient. It can impact mental health and significantly decrease quality of life [1,2].

According to the World Health Organization (WHO), it is estimated that around 50 million people worldwide have epilepsy, with an annual diagnosis rate of approximately five million people [3]. In the United States, the Center for Surveillance, Epidemiology, and Laboratory Services estimated that in 2010 2.3 million adults had active epilepsy, and that this number had increased to three million adults by 2015 [4]. From January to September 2019, patients undergoing assessment or treatment for epilepsy accounted for 2,998,000 consultations in the medical units of the Mexican Institute of Social Security (IMSS) [5]. In [6], through the analysis of six epilepsy studies in Mexico, the prevalence rate was found to be between 3.9 and 42.2 cases per thousand inhabitants. The prevalence varied significantly by location.

The electroencephalogram (EEG) is one of the main tools used for the diagnosis of epilepsy. Manually analyzing EEG recordings is very time consuming; therefore, using computational tools for their analysis and characterization improves the diagnostic process [7]. Several of the computational tools used to analyze EEG recordings are derived from the artificial intelligence field, including machine learning (ML) and deep learning (DL).

On the one hand, ML has been widely applied in epilepsy for seizure detection, differentiation of seizure states, and localization of seizure foci [8]. Conversely, the use of DL techniques in epilepsy has increased regarding classification and prediction tasks [9]. As mentioned by [7], these DL techniques are usually reliant on nontransparent models.

In interesting work presented by [10], the authors proposed novel representations, namely, the unigram ordinal pattern (UniOP) and bigram ordinal pattern (BiOP), to capture underlying dynamics in EEG time series for seizure detection. Their approach demonstrated high accuracy in discriminating between healthy and seizure states, outperforming existing methods. A recent work [11] employed a combination of Variable-Frequency Complex Demodulation (VFCDM) and Convolutional Neural Networks (CNN) to discriminate between health, interictal, and ictal states using electroencephalogram (EEG) data then evaluating CNN performance through leave-one-subject-out cross-validation (LOSO CV), achieving consistently high accuracy rates between healthy and epileptic states.

Unfortunately, it is difficult to explain the decisions of a nontransparent model, and, as argued in [12], explanations of the predictions are necessary to justify the reliability of the models. A solution to this problem can be found in the explainable artificial intelligence (XAI) field, which aims to produce justifications that facilitate the comprehension of a model’s functioning and of the rationale behind its decisions, allowing end users to trust the model [13].

In the state of the art, several studies have explored the utilization of XAI for the analysis of EEG signals and the detection of seizures. In [14], a deep neural network for seizure detection was designed. The model was subjected to adversarial training in order to acquire seizure representations from EEG signals. Additionally, an attention mechanism was implemented to assess the significance of individual EEG channels.

In [15], the connectivity characteristics of EEG signals were estimated, then a set of neural networks was trained to detect seizures. The activation values of the neurons in the classifiers were utilized to estimate the relevance of the characteristics. The findings indicated that the relevance values varied for each subject.

In [16], the authors discussed the application of DL algorithms for the diagnosis of epilepsy. In addition, the use of XAI to explain the model’s decisions was explored. Among the various XAI techniques evaluated, only attention pooling could extract the most significant segments from the signals. However, it was suggested that epileptographic patterns may be too complex to be captured using attention pooling.

Other XAI techniques, such as Shapley Additive Explanations (SHAP) [17], have been employed for different tasks. One such task the classification of the pre-ictal and inter-ictal phases, as described in [18], where the authors used SHAP to assess each EEG channel’s significance and demonstrated how this significance varied over time. Another task for which SHAP has been utilized was presented in [19], where the detection of epileptic seizures from time–frequency domain transformations of EEG was studied using neural networks. Here, the task for SHAP was to visually identify the frequencies that contributed most to the classification.

In [20], the authors proposed a system which uses a Bi-LSTM network for classification of normal and abnormal signals caused by epilepsy and the Layerwise Relevance Propagation (LRP) XAI method to explain the predictions of the network. The LRP method generates a relevance vector for the test input vector. The authors reported that these relevance values indicate the contribution of each datapoint of a signal, helping to classify signals into a particular class.

In another work, SHAP formed part of a methodology known as XAI4EEG developed for the detection of seizures and explanation of the model’s decisions [21]. This technique consists of extracting the time and frequency features of EEG signals for classification by two convolutional neural networks, with SHAP implemented for explanation generation.

In [22], the authors performed minor signal processing steps such as filtering, and used the discrete wavelet transform (DWT) to decompose the EEG signals and extract various eigenvalue features of the statistical time domain (STD) as linear and Fractal Dimension-based Nonlinear (FD-NL) features. Following this feature extraction step, the optimal features were identified through correlation coefficients with p-value and distance correlation analysis and classified using a Bagged Tree-Based Classifer (BTBC), followed by SHAP to provide the explanations.

Based on the understanding that DL models can identify patterns in epileptic EEG signals, are these patterns alone helpful? In addition, if we add transparency using XAI, could the explanations help to identify ictal EEG patterns?

The main objective of this work is to evaluate the utility of explanations generated by XAI techniques such as SHAP and Local Interpretable Model-Agnostic Explanations (LIME) in identifying epileptiform patterns in EEG signals. The aim is to determine whether these explanations can enhance the understanding of deep learning (DL) models and assist in identifying ictal patterns in EEG signals. To achieve this, three EEG databases and a state-of-the-art DL model are utilized to evaluate the models’ performance under different training conditions. Moreover, EEG features are computed for each channel, and the Spearman’s rank correlation coefficient between these features and the importance values generated by XAI techniques is assessed. In summary, this study aims to highlight the complexity involved in identifying ictal patterns from DL models and to explore the role of transparency provided by XAI techniques in this process.

## 2. Materials and Methods

This section describes the methods used for training the models, implementing XAI, and estimating the features. A general methodological overview is shown in Figure 1.

### 2.1. Datasets

The CHB-MIT Scalp EEG database was presented in [23] and is available in a repository [24]. It consists of surface electroencephalograms of 23 pediatric patients with epilepsy: five males (ages 3–22) and 17 females (ages 1.5–19). The patients were undergoing medication withdrawal for epilepsy surgery evaluation. One subject was recorded at an interval of 1.5 years; these recordings are considered as two different cases. There was no gender or age information for one subject. The EEG signals were sampled at a frequency of 256 Hz. The electrodes were placed following the 10/20 system. Most of the recordings contain channels of the longitudinal bipolar montage. As mentioned in [23], the dataset was mostly segmented into 1 h recordings. The recording device is not mentioned.

The Siena Scalp EEG database was presented in [25] and is available at [26]. This dataset contains surface EEG recordings from 14 adult patients: nine males (ages 36–71) and five females (ages 20–58). The recordings have a duration between 1 h and 13 h. The signals were recorded with a sampling frequency of 512 Hz. The electrodes were placed according to the 10/20 system. The channels were monopolar. EB Neuro and Natus Quantum LTM amplifiers were utilized for data acquisition. The subjects were monitored using a video scalp EEG and were asked to stay in bed most of the time.

The TUH EEG Seizure Corpus (TUSZ) was presented in [27,28]. It consists of surface EEG recordings from 315 subjects obtained over 822 sessions. Only 280 sessions contain a seizure. The gender composition is 153 males and 162 females. The dataset includes both pediatric and adult patients. Most EEG recordings have a duration between 0 and 30 min. The sampling frequency varied per patient (250 Hz, 256 Hz, 400 Hz, and 512 Hz). The channels were monopolar. The recording devices are not mentioned. Due to the breadth of this database, the recording protocols, ages, and recording durations are varied (refer to [27] for detailed information).

### 2.2. Data Pre-Processing

EEG recordings were discarded based on the following criteria:They did not contain ictal activity;The channel list was different from that of the rest of the recordings;The sampling frequency was different from that of the rest of the recordings;There was only a single recording for each patient.

Another consideration for this study was the use of bipolar longitudinal montages. This choice was motivated by the prevalence of databases presenting recordings in this format and the advantage of transforming monopolar montages into bipolar rather than vice versa.

Due to the large amount of data, only a subset of patients was selected from the TUSZ dataset. The number of patients, channels, and seizure types is detailed in Table 1. Finally, a Notch filter was applied to the EEG windows to remove the power line frequency. Moreover, a second-order high-pass Butterworth filter was used to remove frequencies under 0.5 Hz.

### 2.3. Dataset Segmentation, Training, and Testing

According to the methodology described in [29], the EEG signals were segmented into windows of 2 s. In order to ensure that both classes, ictal and non-ictal, were represented equally during training and validation, a two-step balancing process was implemented: the first step consisted of oversampling the ictal epochs with three different overlap ratios (80%, 70%, and 50%), while the second step involved subsampling from the larger class. While evaluating the DL models, the windows were created without overlap to avoid utilizing known data, and both classes were balanced. From the training set, 20% was used as the validation set.

### 2.4. Deep Learning Models

A literature search was conducted to select the DL algorithm. The search criteria were as follows:Papers were included in PubMed and Clarivate;Papers were published after 2015;Papers were available via open access;Papers described bi-class classification (ictal and non-ictal);Papers described the use of raw EEG data;Papers described the implementation of a deep learning model;Papers described patient-specific models;Papers described the use of performance metrics.

After reviewing the papers meeting the above criteria, the model presented by [30] was selected. Table 2 displays the performance and characteristics of the model. In [30], two approaches were tested: segment-based and event-based. The segment-based approach was relevant to the present research.

The model is a Siamese convolutional neural network (CNN), and is illustrated in Figure 2. The convolutional layers utilize ReLU as the activation function, while Softmax is applied to the output layer for activation. A dropout layer is appended between the fully connected layers, with a dropout rate equal to 0.25. Throughout the remainder of this document, this model is referred to as Wang_1d. We used different parameters than those used in [30]. The Adam algorithm was employed as the optimizer, using the following parameters: $\alpha =0.001,\beta_{1}=0.9,\beta_{2}=0.999$, and $\epsilon =1\times 10^{-7}$. Binary cross-entropy was used as the loss function and accuracy as the evaluation metric. The number of epochs and the batch size were both 100.

### 2.5. Model Evaluation

The models were trained and evaluated using an intra-patient approach. A similar approach to leave-one-out cross-validation was implemented for model evaluation. A neural network was trained *k* times using a specific overlap rate, with *k* representing the number of recordings per patient. In the first iteration, k−1 recordings were utilized for training and the remaining recordings were used for evaluation.

The specificity, sensitivity, accuracy, AUC–ROC, and F1-score were calculated for each iteration. Specificity and sensitivity were used to measure the capacity of the model to detect true positives and true negatives. Accuracy and F1-score provide a general measure of performance, and are commonly computed for seizure detection models, while AUC-ROC helps to visualize the separability between classes.

### 2.6. Feature Computation

The EEG windows were separated into the following three sub-bands: low frequencies (<12 Hz), beta (12–25 Hz), and gamma (>25 Hz). Second-order Butterworth filters were used for band separation.

A set of 14 features that simplified the behavior of segmented bipolar EEG signals by capturing some of their key characteristics was computed; these are mentioned in Table 3. Features were computed for each sub-band separately, and primal EEG epochs were considered during computation. The total number of features was 14 features× 4 bands=56. The selection of features was based on previous research [31]. Although an exhaustive analysis was not performed, the selected features have been utilized in previous studies of epilepsy.

### 2.7. Explainable Artificial Intelligence

In this work, the Shapley Additive Explanations (SHAP) and Local Interpretable Model-Agnostic Explanations (LIME) techniques were chosen to assess the model’s explainability.

#### 2.7.1. Shapley Additive Explanations

SHAP is an approach for interpreting the predictions made by ML models [17]. To evaluate an instance, each attribute is assigned a SHAP value, which indicates the relative importance of the attribute to the model’s decision-making process. The formal definition is as follows:(1)$$\phi_{i}\left(f,x\right)\;=\;\sum_{z^{\prime }\subseteq x^{\prime }}\frac{|z^{\prime }\left|(M-\right|z^{\prime }\left|-1\right)!}{M!}\left[f_{x}\left(z^{\prime }\right)-f_{x}\left(z^{\prime }\setminus i\right)\right]$$
where *x* is the instance to be explained, *f* is the model, *i* is the feature to be evaluated, and *M* is the number of features. Additionally, $x^{\prime }$ contains all possible perturbations of *x*.

In this work, we utilized the PartitionSHAP algorithm [38], which is a component of the software package introduced by the authors of [17]. This algorithm allows for the computation of importance values by evaluating a group or coalition of features. Consequently, the features of a given coalition receive the same SHAP value.

PartionSHAP computes Shapley values using a hierarchical approach that defines coalitions and returns Owen values [39]. The hierarchy depth allows the coalition size to be determined. We applied PartitionSHAP solely to the classification model exhibiting the best overall performance (refer to Section 3 for further details). Subsequently, we estimated the SHAP values for those instances correctly classified as ictal.

The depth of the PartitionSHAP hierarchy was tailored such that each coalition corresponded to a single channel of the EEG window (see Figure 3). Consequently, a single importance value was computed per channel. Two SHAP values were obtained for specific channel pairs, as the PartitionSHAP hierarchy operates through powers of 2. In such cases, the importance of these coalitions was determined as the mean value of both SHAP values.

#### 2.7.2. Local Interpretable Model-Agnostic Explanations

LIME is an approach for explaining the predictions of any classification model by approximating it locally using an interpretable model [40]. An interpretable model should provide an understanding of both the input variables and the response.

The formal definition of LIME’s explanation is as follows:(2)$$\xi \left(x\right)=\arg\;\min_{g\in G}L\left(f,g,\pi_{x}\right)+\Omega \left(g\right)$$
where *x* is the instance to be explained, ξ is the instance explanation, *g* is a potentially interpretable model such as a linear model or decision tree, and *f* is the classification model. The function *L* measures the approximation of *g* to *f* in the locality defined by $\pi_{x}$. The complexity of *g* is measured by Ω(g); this parameter is related to the complexity of the model *g*.

To approximate $L(f,g,\pi_{x})$, a set of instances around *x* is sampled. The sampling is performed by perturbing the features of *x*, with the nearer instances having higher weights. Later, the model *g* is adjusted on the basis of the perturbed dataset by considering the instance weights.

In this work, we utilized the LimeImageExplainer algorithm introduced by the authors of [40]. This method considers the neighbors’ features as superpixels. Therefore, a superpixel’s members receive the same importance value. Additionally, the algorithm uses the Euclidean distance to weight the perturbed instances and Ridge regression as the interpretable model. The adjusted regressor coefficients are considered as the importance values.

The method was adjusted to evaluate each channel as a group (see Figure 4); therefore, each channel received a unique importance value. Similar to PartitionSHAP, LIME was solely applied to the classification model exhibiting the best overall performance (refer to Section 3 for further details).

### 2.8. Correlation Computation

Every channel has an importance value and a corresponding set of features. For a given feature (e.g., the complexity of the beta band), we computed the Spearman’s rank correlation coefficient [41] between the feature and the importance values. This analysis was repeated for every feature and patient. During the correlation experiments, the non-ictal windows were not considered, as this research aimed to understand the behavior of the ictal stage.

### 2.9. Computing Hardware

The experiments were conducted on two devices with the following technical specifications: a computer equipped with an Intel Core i7 processor, 12 GB of RAM, and Ubuntu 18.04, and a server equipped with an Intel Xeon Gold processor, 125 GB of RAM, and an NVIDIA Tesla P100, Python 3.7 [42] was employed to code all experiments, along with various open-access Python libraries.

## 3. Results

In this section, the results of our experiments are presented. First, we present the model’s performance when trained with varying percentages of overlapping windows for each of the EEG datasets. Finally, we examine the correlation between the importance values and the EEG features.

### 3.1. Model Performance

This analysis aimed to visualize the variations in performance for a deep learning neural network subjected to various conditions. The varied conditions were as follows. Overlapping was applied to the training set and the EEG dataset used for training; the window overlapping on the training set functioned as a data augmentation technique to help balance the classes. In order to avoid biasing the results, the test set was not augmented.

Table 4 presents the results of the classification models. The metrics were averaged across patients. The performance on the CHB-MIT database was superior to that on the Siena and TUSZ datasets. The sensitivity and specificity were greater than 0.82 and 0.60, respectively.

Considering that the test set is balanced and that we are modeling a bi-class problem, accuracy is a good metric for evaluating overall performance. In this sense, the highest accuracy was obtained using a 80% overlap for each dataset: 0.84 (CHB-MIT), 0.73 (Siena), and 0.64 (TUSZ). As expected, this may imply that using more instances used during training results in better modeling. Note that none of the individual test sets were augmented. Additionally, even when the accuracy metric was calculated, sensitivity and specificity were computed as well in order to obtain a deeper view of the ictal and non-ictal predictions. It should be noted that the model’s sensitivity was always greater than its specificity, which in our case implies that the model performed better on identifying the target class (ictal). The difference between the sensitivity and specificity was considerable for the Siena dataset (50% and 70% overlap rate, respectively).

The specificity rose when the overlap was increased; on the other hand, the sensitivity tended to decrease. Generally speaking, the model’s accuracy was greater as the overlap rate increased. Contrary to the Siena and CHB-MIT datasets, on the TUSZ dataset these effects might be a consequence of the mixture of participants, particularly the mixture of adult and pediatric subjects. It is important to consider that the results in Table 4 correspond to the average of all participants per dataset. The largest F1-score was displayed when using an overlap rate of 80%:0.83 (CHB-MIT), 0.71 (Siena), and 0.62 (TUSZ). Unlike the accuracy, it did not follow an ascending trend. Finally, the AUC-ROC score was over 0.85 for the CHB-MIT database, while TUSZ returned the lowest score (0.70, 50% overlap). The scores for the CHB-MIT and Siena datasets did not display an ascending trend.

To depict the individual prediction behavior of the models, Figure 5 displays the model performance for each EEG dataset. It includes the metrics for all patients (the number of subjects can be observed in Table 1). A 70% overlap was used to generate this chart. The median of the CHB-MIT results was higher than that of the other two datasets; this situation applied to four metrics, specificity, accuracy, F1-score and AUC-ROC. Therefore, it can be intuited that the Wang_1d model performed best when using the CHB-MIT dataset.

Moreover, the TUSZ-related values spanned an extensive range, meaning that the model was able to achieve good performance for some patients while showing poor results for others. As displayed in Table 4, the models tended to be more sensitive than specific. As mentioned before, it is important to consider that the TUSZ dataset contains both pediatric and adult patients; thus, such heterogeneity may be reflected in the dispersion of the prediction performances.

Figure 6 shows the model accuracy when varying the overlap rate. As stated, the overlap was only applied to generate the training dataset, not the evaluation one. When using the CHB-MIT and Siena datasets, it was noted that the overall accuracy increased when the overlap increased. On the other hand, the TUSZ-related models did not show this behavior. Again, the TUSZ-related ranges were the largest among the three datasets. It is noticeable that, visually speaking, there is no effect when using different window overlaps and the accuracy is lower among all datasets.

The Friedman test [41] returned a *p*-value $=7.48\times 10^{-5}$ for the CHB-MIT dataset and 0.02 for the Siena dataset, indicating that increasing the overlap could increase the model’s performance. The previous statement is supported by Figure 6 and Table 4. The Friedman test applied to the TUSZ accuracy values returned a *p*-value =0.54.

Figure 7 displays the model accuracy for each patient. A 70% overlap was used to estimate the performance. Each point denotes a patient. The variable used to perform the comparison was the seizure type. When a patient suffered from several seizure types, the most common type was considered.

Figure 7a,b displays the performance for subjects belonging to the Siena and TUSZ datasets, respectively. As per the charts, there was no clear relation between seizure type and model performance. The CHB-MIT values were not included, as the type of seizure was not explicitly mentioned for each patient.

### 3.2. Correlation between Importance Values and EEG Features

Considering that the greatest accuracy was achieved when using an 80% overlap, these models were used to estimate the importance values and correlation coefficients.

Figure 8, Figure 9 and Figure 10 show the distribution of the Spearman’s rank correlation coefficients. The histograms are individually displayed for the SHAP and LIME experiments. The coefficients within the blue dashed lines can be interpreted as negligible correlations. The coefficients between the red and blue dashed lines denote a low correlation (see Table 5). It is essential to consider that the Spearman’s rank correlation coefficient measures the strength of a monotonic relationship between two variables.

The Spearman’s rank correlation coefficients estimated using the CHB-MIT dataset are displayed in Figure 8. The chart shows that there was no moderate or high correlation between the SHAP importance values and EEG features. The same was true for the LIME values. The vast majority of coefficients were near zero. A number of the experiments displayed a low positive/negative monotonic relation.

The results in Figure 9 are similar to those in Figure 8. Most coefficients fall within the blue dashed lines; these correlations are negligible. This analysis applies to both the SHAP and LIME experiments. In Figure 9a, a number of of the coefficients indicate a moderate positive correlation; these are discussed later.

Notwithstanding this, Figure 10 displays similar behavior to that in Figure 8 and Figure 9, as a more considerable number of the Spearman’s rank correlation coefficients surpassed the 0.50 and 0.70 thresholds (moderate and high correlation, respectively).

Five of the six patients were part of the TUSZ dataset. The features that had a moderate/strong monotonic relation with the XAI explanations were diverse, including STD, MAD IR, RMS, and Range. Most features were estimated for the beta band (12–25 Hz).

Although the experiments returned a moderate correlation, the model accuracy was unsatisfactory for several patients (PN05, 1027 and 6904). Therefore, a moderately strong correlation does not imply a proficient model. The only experiment returning a high correlation corresponded to patient 4456, where, remarkably, the model accuracy was 0.96. The most common seizure type for the previous patient was gnsz.

Table 6 details the patients for whom the experiments displayed a moderate/high correlation between the importance values and EEG features. Regarding the *p*-value, the null hypothesis is that the samples have no ordinal correlation; the alternative hypothesis is that the correlation is nonzero.

## 4. Discussion

This section presents a discussion of the results and their interpretation. In addition, the limitations of the present work and future research opportunities are addressed.

A state-of-the-art DL model was evaluated in this study, specifically, the one-dimensional convolutional neural network for seizure onset detection presented in [30]. Despite aiming to select models that demonstrated high performance and rigor in their evaluation, it should be acknowledged that the search for models could have been more comprehensive.

In [30], two databases were employed, namely, CHB-MIT [23] and SWEC-ETHZ iEEG [44], with the former being surface and the latter intracranial. Furthermore, the models were trained using an intra-patient approach. The results reported in [30] using the CHB-MIT data showed mean sensitivity, specificity, and accuracy values of 0.88, 0.99, and 0.99, respectively. In contrast, our study utilizing the same EEG dataset (see Table 4) reported maximum sensitivity, specificity, and accuracy of 0.84, 0.84, and 0.84, respectively, with significantly lower specificity. It is essential to consider the differences in data formation between [30] and our work, such as the use of short-duration seizures, avoidance of seizure concatenation, and overlapping rates.

A brief overview of similar studies conducted using comparable methodologies is presented next. In [45], a sensitivity of 0.976 was reported for intra-patient models; this was achieved by applying a convolutional neural network in conjunction with a long short-term memory network. In [46], the authors employed inter-patient models using only the CHB-MIT database, and reported mean values of 0.90, 0.91, and 0.98 for sensitivity, specificity, and accuracy, respectively. In [47], the authors trained patient-specific convolutional neural networks and reported values of 0.90 and 0.98 for sensitivity and the area under the curve, respectively. In [48], the authors utilized a neural network called ScoreNet, achieving a sensitivity of 0.765 and specificity of 0.999. Finally, an accuracy of 0.805 was obtained by [14], who trained inter-patient models using an adversarial neural network. Among the mentioned works, ref. [45,46,47,48] used the CHB-MIT database, while [14] used the TUH EEG Seizure Corpus. However, there are other works that have reported better performance in terms of accuracy, such as [11] for advanced epilepsy detection by applying VFCDM and CNN, who achieved consistently high accuracy rates between healthy and epileptic states. However, their results were obtained for the Bonn database, which contains only five patients with epilepsy. In our work exploring the importance of EEG channels in detecting epileptic seizures, we have identified that database and training conditions can affect classifier performance.

Regarding the work presented in [10], the authors selected three EEG datasets, one of which was the publicly available Bonn dataset, to demonstrate that their method outperformed several state-of-the-art methods. A second dataset was used to show the good generalization ability of their proposed method, and a third dataset to demonstrate that their method is suitable for large-scale datasets. In our work, we selected three EEG datasets to train and evaluate the models under different conditions of overlap between EEG data windows.

One of the goals of the present research was to understand how the training conditions can impact a model’s performance. First, the results showed that a model’s sensitivity and specificity can vary noticeably based on the EEG dataset. For example, the overall specificity for CHB-MIT models was more significant than for the rest of the models (see Figure 5). Accordingly, any ML model used to detect seizures should be evaluated across diverse EEG datasets.

The differences in performance can be explained by the differences between the EEG datasets, such as the EEG channels, sampling frequency, patient demographics, and epilepsy characteristics. Second, it was observed that an overlap during training impacts the model’s performance. During this research, the overlap was applied to the ictal instances to address class imbalances. In addition, the largest class (non-ictal) was subsampled to generate a balanced dataset.

Our results showed that the more significant the overlap applied to the ictal class, the higher the accuracy. It should be noted that a significant overlap implies that the training dataset size has increased. An exception is the TUSZ dataset; Table 4 shows an increase in overall accuracy for the TUSZ models, although the increase is not significant.

Our work can be compared to similar state-of-the-art works with the primary objective of automatically detecting seizures by applying machine learning methods to EEG signals and incorporating explainable artificial intelligence techniques to enhance the interpretability of the models used in seizure detection. First, we have the work presented in [20]; the authors proposed a system which uses a Bi-LSTM network to classify normal and abnormal signals caused by epilepsy, achieving an accuracy of 87.25% on the Bonn dataset. They used the Layerwise Relevance Propagation (LRP) XAI method to explain the predictions of their network. LRP generates a relevance vector containing relevance values to indicate the contribution of a signal in particular class. However, as stated by the authors, many points in the relevance vector were missed; thus, this implementation of the LRP method requires further improvement to generate more accurate results. In [22], the main difference with our work is in the feature extraction step. They reported an accuracy of 99.6% on the Bonn dataset, and their explanations were over the features rather than the signal morphology (time series). In our work, we chose a state-of-the-art deep learning model for seizure detection and three EEG databases. The developed models were trained and evaluated under different conditions and the classifiers with the best performance were selected. SHAP and LIME were then employed to estimate the importance value of each EEG channel. To measure the similarity between the explanations and the epileptic signals, we computed the Spearman’s rank correlation coefficient between the EEG features of epileptic signals (time series) and the importance values. Another important difference between these works are the data sources. In [20,22], the authors used an open access dataset published by researchers at Bonn University containing intracranial EEG recordings from a total number of ten subjects, of whom five were healthy volunteers and the other five were epilepsy patients. Compared to our work, in which three databases were tested, previous works have used a limited dataset in order to evaluate their algorithms. Furthermore, studies employing the Bonn dataset used intracranial instead of scalp EEG recordings.

The interpretability of classification models used in the medical field is crucial, as mentioned by [12,49]; although the number of XAI algorithms is significant, not all algorithms can be applied to time series. Compared to other fields, the interpretation of time series in this field is usually not intuitive, and requires domain knowledge [13,50]. In the present work, SHAP and LIME, which are both agnostic and local methods, were applied. SHAP has previously been used for EEG signals; selected use cases are displayed in Table 7.

In light of the capacity of XAI methods to identify information relevant to the classifier, it may be possible to identify discriminating ictal patterns in high-importance regions (e.g., large SHAP values would match large STD values). As XAI aims to provide transparency, understandable ictal patterns would be preferable.

In line with the above, the Spearman’s rank correlation coefficient between the XAI-generated importance values and the EEG features was computed. As previously stated, this coefficient measures the strength of a monotonic relationship.

As deduced from the results, the strength of the relationship was low and negligible even for high-performing models. The most remarkable cases are described in Table 6. It was observed that experiments resulting in a moderate correlation did not present high accuracy; therefore, a larger correlation coefficient does not imply better performance. The only exception was patient 4456 (TUSZ dataset).

To the best of the authors’ knowledge, there is no similar study in the state of the art; hence, the comparison was complex. However, the difficulties in obtaining epileptic EEG patterns have been described in previous works, e.g., [15,18]. It is relevant to mention that while several previously published works have included XAI to increase the transparency, they did not apply a pattern/explanation evaluation stage.

In addition to the above, the following limitations must be considered: the applied XAI methods, the small set of EEG features, and the chosen classification model. Although more extensive experiments could be performed to search for correlation patterns, alternative approaches must be considered. These are outlined below.

In [55], the classification of seizure and non-seizure states was performed. Random forest showed the best accuracy among the evaluated classification models. Bidirectional network graphs and the lifespan of homology classes were computed to characterize the EEG windows. SHAP was implemented to provide model transparency. The use of EEG features as model inputs provided an initial foundation for transparency, contrary to the raw EEG signals.

In [56], the classification of eight seizure types was addressed. A deep neural network was used for classification and raw EEG windows were used as model inputs. SHAP and topographic maps were applied for transparency. Notably, network activation was used to plot the topographic maps. This technique adds a spatial feature to the XAI explanations, which may indeed be helpful for medical staff.

Additionally, seizure prediction was addressed in [57]. A list of univariate linear features was estimated from EEG recordings. Support vector machine, logistic regression, and CNN were applied during classification. A diverse set of techniques was used to provide explainability, e.g., SHAP, LIME and partial dependency plots. Importantly, the explanations were evaluated by humans (data scientists and clinicians).

Finally, there are a number of limitations in the present work. First, even though a deeper literature revision was performed, modeling the three datasets using several architectures is required to measure the effect of architecture variations. Second, it would be interesting to group the information by several different conditions (for example, on the basis of seizure type, sex, and age, among others) in order to evaluate their effects on detection of ictal patterns. Third, as the use of the Spearman correlation constrains the search to monotonic relations, different nonlinear metrics should be used.

## 5. Conclusions

The main contribution of this work is the evaluation of the Spearman’s rank correlation coefficient between the features of EEG signals and XAI explanations to identify ictal patterns. It was observed that the EEG dataset impacted the performance of the classification models. Additionally, significant overlaps during training may increase model performance. Our results indicate a negligible and low correlation coefficient between the evaluated features and the LIME/SHAP values, although a few exceptions were observed.

For future work, it is recommended to perform a tradeoff analysis considering the model’s performance and explainability as variables. Clinicians should participate in the explainability evaluation.

## Figures and Tables

**Figure 1 brainsci-14-00306-f001:**
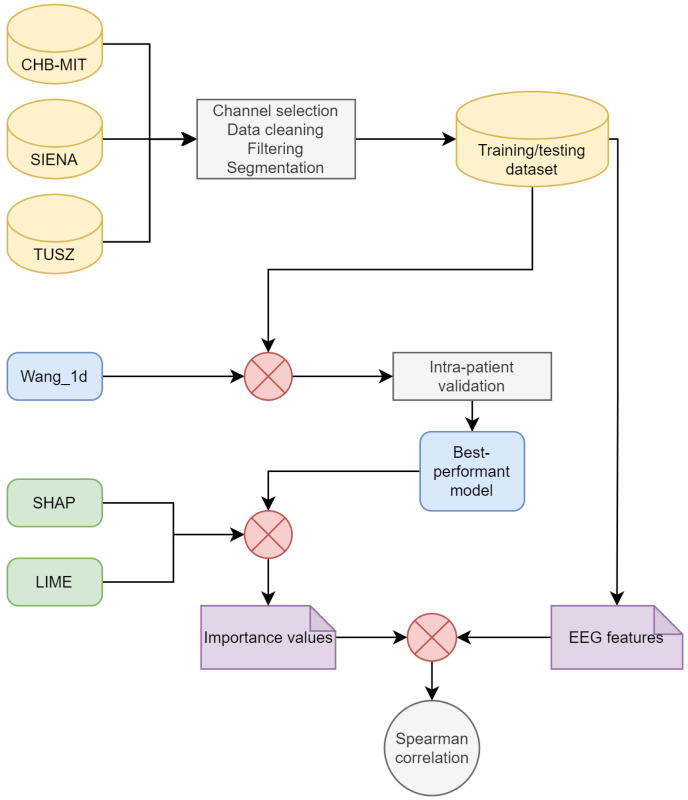
Diagram of the methodology.

**Figure 2 brainsci-14-00306-f002:**
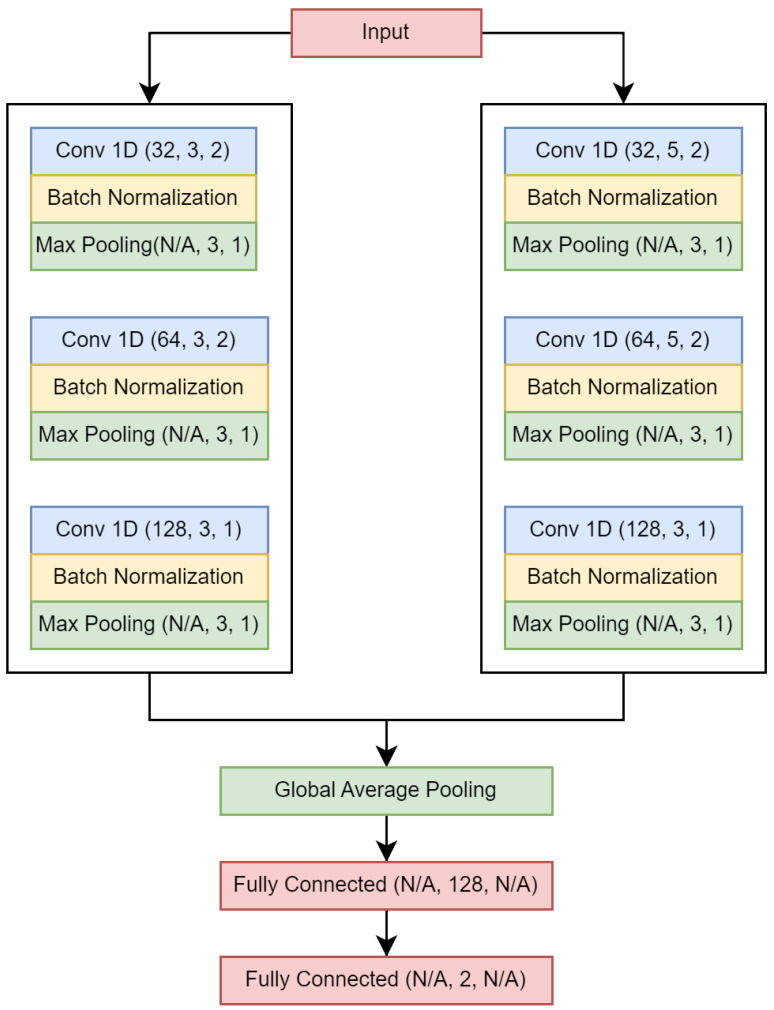
Wang_1d neural network. Values in parentheses are as follows: (number of filters, kernel size/number of neurons, stride).

**Figure 3 brainsci-14-00306-f003:**
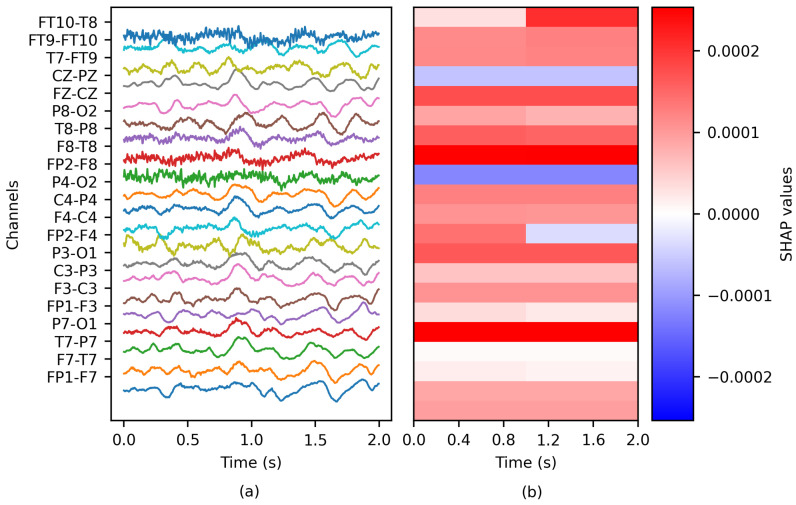
The application of SHAP to an EEG epoch: (**a**) the ictal EEG window of patient chb01 and (**b**) the matrix of SHAP values.

**Figure 4 brainsci-14-00306-f004:**
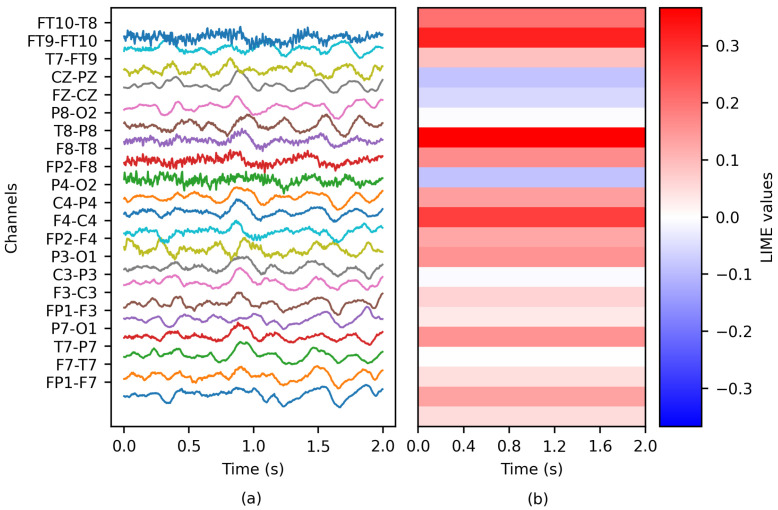
The application of LIME to an EEG epoch: (**a**) the ictal EEG window of patient chb01 and (**b**) the matrix of LIME values.

**Figure 5 brainsci-14-00306-f005:**
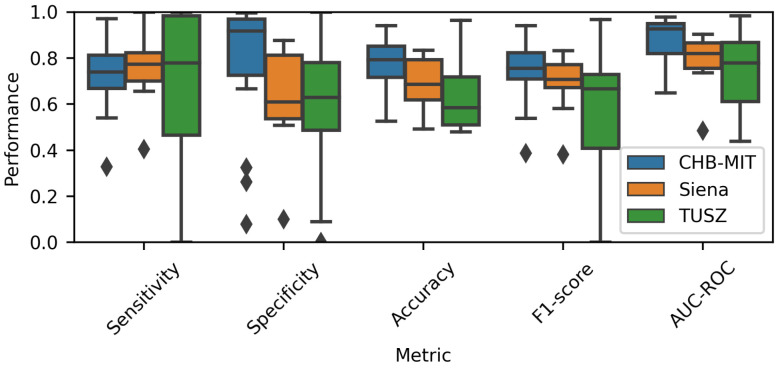
Performance comparison for each EEG database. A 70% overlap was used to estimate the metrics.

**Figure 6 brainsci-14-00306-f006:**
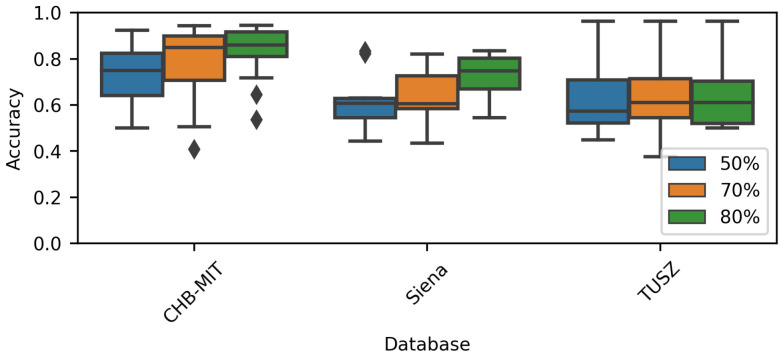
Performance comparison for each overlap rate.

**Figure 7 brainsci-14-00306-f007:**
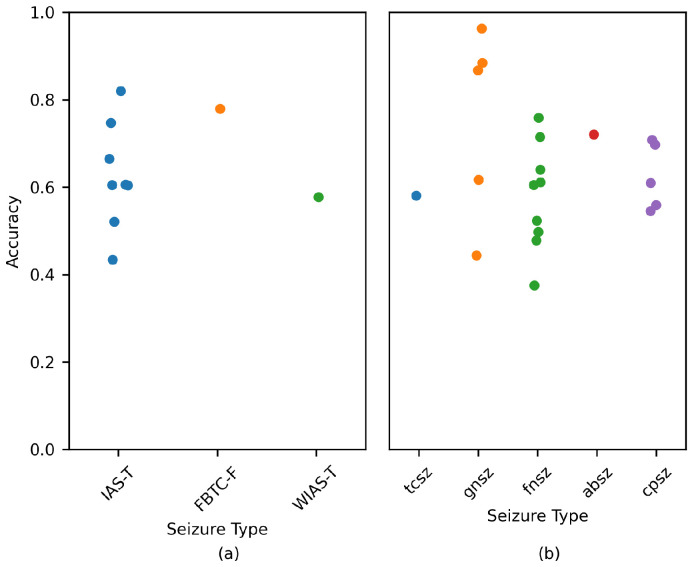
Performance comparison for each patient based on the most common seizure type. Data are displayed for the (**a**) Siena and (**b**) TUSZ datasets. A 70% overlap was used to estimate the metrics. Each circle denotes a patient.

**Figure 8 brainsci-14-00306-f008:**
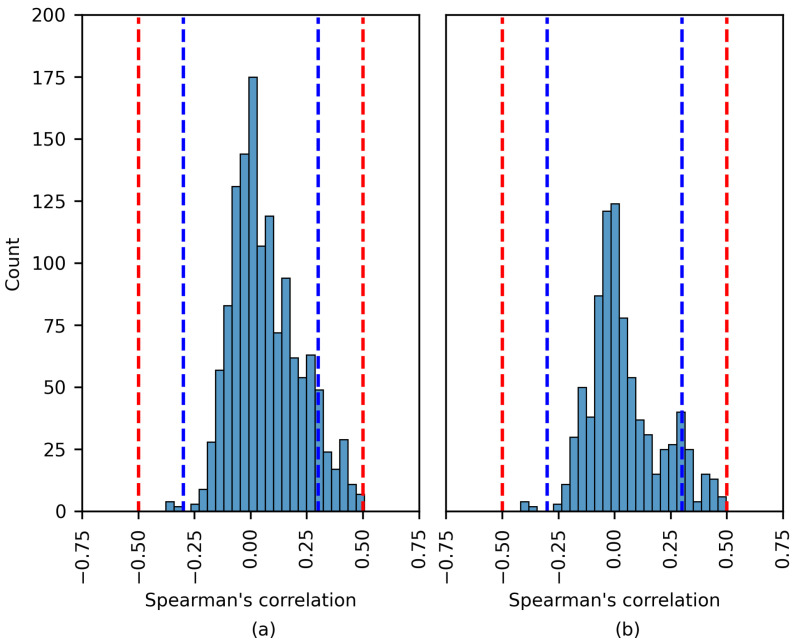
The distribution of the Spearman´s rank correlation coefficients estimated for the CHB-MIT dataset. The range between the blue dashed lines corresponds to a negligible correlation, while the range between the blue and red dashed lines corresponds to a low correlation. (**a**) SHAP values; (**b**) LIME values.

**Figure 9 brainsci-14-00306-f009:**
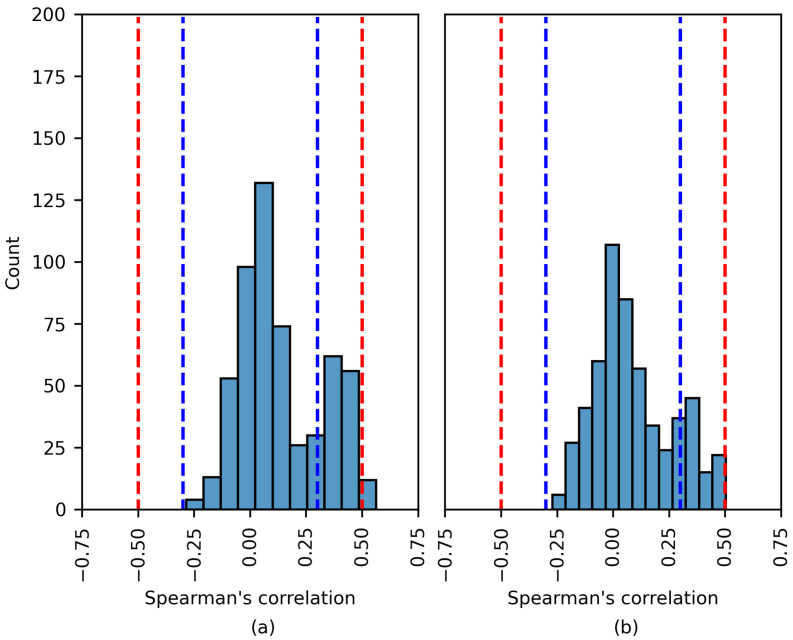
The distribution of the Spearman’s rank correlation coefficients estimated for the Siena dataset. The range between the blue dashed lines corresponds to a negligible correlation, while the range between the blue and red dashed lines corresponds to a low correlation. (**a**) SHAP values; (**b**) LIME values.

**Figure 10 brainsci-14-00306-f010:**
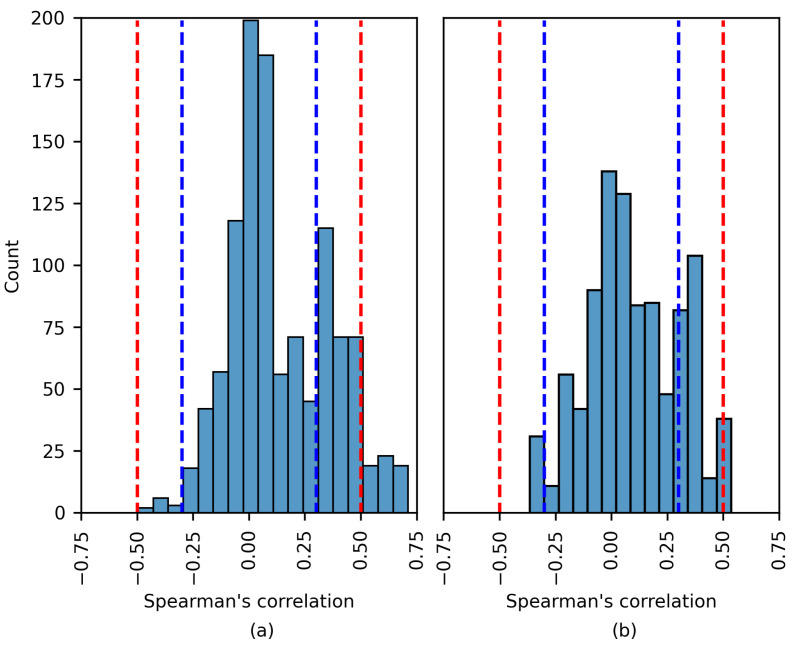
The distribution of the Spearman’s rank correlation coefficients estimated for the TUSZ dataset. The range between the blue dashed lines corresponds to a negligible correlation, while the range between the blue and red dashed lines corresponds to a low correlation. (**a**) SHAP values; (**b**) LIME values.

**Table 1 brainsci-14-00306-t001:** Details of the datasets after preprocessing.

Database	Patients	Seizure Types	Source Montage Category	Channels after Pre-Processing
CHB-MIT Scalp EEG Database	24	Not specified	Bipolar	Fp1-F7, F7-T7, T7-P7, P7-O1, Fp1-F3, F3-C3, C3-P3, P3-O1, FP2-F4, F4-C4, C4-P4, P4-O2, FP2-F8, F8-T8, T8-P8, P8-O2, FZ-CZ, CZ-PZ, T7-FT9, FT9-FT10, FT10-T8
Siena Scalp EEG Database	10	Focal onset impaired awareness (IAS), focal onset without impaired awareness (WIAS), focal to bilateral tonic–clonic (FBTC)	Referential	Fp1-F7, F7-T3, T3-T5, T5-O1, Fp1-F3, F3-C3, C3-P3, P3-O1, Fp2-F4, F4-C4, C4-P4, P4-O2, Fp2-F8, F8-T4, T4-T6, T6-O2, Fz-Cz, Cz-Pz
TUH EEG Seizure Corpus	21	Tonic–clonic (tcsz), focal non-specific (fnsz), generalized non-specific (gnsz), absence seizure (absz), complex-partial seizure (cpsz)	Referential	Fp1-F7, F7-T3, T3-T5, T5-O1, Fp1-F3, F3-C3, C3-P3, P3-O1, Fp2-F4, F4-C4, C4-P4, P4-O2, Fp2-F8, F8-T4, T4-T6, T6-O2, Fz-Cz, Cz-Pz

**Table 2 brainsci-14-00306-t002:** Performance and characteristics of the model presented by [30].

Characteristic	Value
Model	CNN
Training type	Patient-specific
Accuracy (CHB-MIT)	0.99
Sensitivity (CHB-MIT)	0.88
Specificity (CHB-MIT)	0.99

**Table 3 brainsci-14-00306-t003:** Estimated EEG features.

Feature Name
Median Frequency (MedFreq) [32]
Complexity [33]
Skewness [34]
Mobility [33]
Kurtosis [34]
Interquartile Range (IR) [34]
Peak Frequency (PkFreq) [35]
Median Absolute Deviation (MAD) [36]
Root Mean Square (RMS) [34]
Sample Entropy (SampEn) [37]
Range [34]
Mean [34]
Number of Zero Crossings (ZC) [33]
Standard Deviation (STD) [34]

**Table 4 brainsci-14-00306-t004:** Mean performance of the classification models.

Model	Overlap (%)	Sensitivity	Specificity	Accuracy	F1-Score	AUC–ROC
		CHB-MIT				
Wang_1d	50	0.83	0.61	0.72	0.73	0.86
	70	0.83	0.77	0.79	0.60	0.88
	80	0.84	0.84	0.84	0.83	0.88
		Siena				
Wang_1d	50	0.84	0.39	0.62	0.68	0.78
	70	0.80	0.53	0.63	0.56	0.79
	80	0.74	0.71	0.73	0.71	0.78
		TUSZ				
Wang_1d	50	0.73	0.49	0.61	0.61	0.70
	70	0.69	0.58	0.63	0.54	0.73
	80	0.69	0.58	0.64	0.62	0.74

**Table 5 brainsci-14-00306-t005:** Interpretation of the size of a correlation coefficient. Taken from [43].

Size of Correlation	Interpretation
0.90 to 1.00 (−0.90 to −1.00)	Very high positive (negative)
0.70 to 0.90 (−0.70 to −0.90)	High positive (negative)
0.50 to 0.70 (−0.50 to −0.70)	Moderate positive (negative)
0.30 to 0.50 (−0.30 to −0.50)	Low positive (negative)
0.00 to 0.30 (−0.00 to −0.30)	Negligible correlation

**Table 6 brainsci-14-00306-t006:** List of patients for whom the experiments displayed a moderate/high correlation between the importance values and EEG features. The ’Top Correlation’ column indicates the three features with the largest correlation coefficients.

Patient	Dataset	Accuracy	XAI Method	Top Correlation
PN05	Siena	0.54	SHAP	Low/STD (0.55, *p*-value $=2.42\times 10^{-48}$)
				Low/MAD (0.55, *p*-value $=2.05\times 10^{-49}$)
				Low/IR (0.55, *p*-value $=2.30\times 10^{-49}$)
1027	TUSZ	0.70	SHAP	Full/IR (0.53, *p*-value =0)
				Full/RMS (0.53, *p*-value =0)
				Low/Range (0.54, *p*-value =0)
4456	TUSZ	0.96	SHAP	Beta/IR (0.70, *p*-value $=9.31\times 10^{-50}$)
				Beta/RMS (0.71, *p*-value $=1.58\times 10^{-50}$)
				Beta/STD (0.71, *p*-value $=1.59\times 10^{-50}$)
6904	TUSZ	0.63	SHAP	Beta/IR (0.66, *p*-value =0)
				Beta/MAD (0.66, *p*-value =0)
				Beta/RMS (0.66, *p*-value =0)
			LIME	Beta/MAD (0.53, *p*-value =0)
				Beta/RMS (0.53, *p*-value =0)
				Beta/STD (0.53, *p*-value =0)
6563	TUSZ	0.79	LIME	Beta/MAD (0.52, *p*-value =0)
				Beta/RMS (0.53, *p*-value =0)
				Beta/STD (0.53, *p*-value =0)

**Table 7 brainsci-14-00306-t007:** XAI methods applied to EEG signals.

Reference	Applied Methods
[51]	Bag of Waves
[52]	Clustered Pattern of Highly Activated Period
[18]	SHAP
[16]	ProtoPNet, Attention Pooling, Layerwise Relevance Propagation, SHAP
[15]	Activation values
[14]	Attention Pooling
[19]	SHAP
[53]	EEG-SHAP
[54]	SHAP, Maximized inputs

## Data Availability

The CHB-MIT Scalp EEG Database is available at https://physionet.org/content/chbmit/1.0.0/ (accessed on 26 January 2023). The Siena Scalp EEG Database is available at https://physionet.org/content/siena-scalp-eeg/1.0.0/ (accessed on 26 January 2023). For access to the TUH EEG Seizure Corpus, refer to [27].

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
