# Peer review of "Evaluation of the Relation between Ictal EEG Features and XAI Explanations"

_brainsci, 2024, doi:10.3390/brainsci14040306_

Round 1
Reviewer 1 Report
Comments and Suggestions for Authors
- The paper has small paragraphs in all the sections: from Introduction to Conclusion. Please correct them.
-
Clearly define the main research novelty and contributions objectives as a last paragraph of introduction of the paper. Which provides a clear understanding of the study's purpose to the readers.
-
Provide a more detailed description of the methodology used for training the deep-learning neural network, including information on data preprocessing, model architecture, hyperparameters, and training/validation procedures.
- Detailed description of the databases is required: Participants information (demographics), gender ratio, Selection criteria, Protocol used, device used for recording, duration of signal recording etc.,
-
Ensure consistency in formatting throughout the document, including tables, figures, and references, to improve readability and presentation quality.
-
Provide more detailed explanations and interpretations of the results, particularly regarding the observed variations in model performance across different EEG datasets and overlap percentages.
-
Expand the discussion of limitations to include potential sources of bias, confounding variables, and other factors that may have influenced the results, and suggest ways to address these limitations in future studies.
-
Provide a more comprehensive comparison with related work in the field, including recent studies on seizure detection using EEG data, to highlight the novelty and significance of the research findings.
- Compare and discuss recent works such as: https://doi.org/10.3390/signals4040045
-
Consider conducting additional statistical analyses, such as hypothesis testing or cross-validation, to validate the significance of the observed correlations between XAI explanations and EEG features.
-
Discuss the choice of evaluation metrics used to assess model performance and consider including additional metrics, such as area under the receiver operating characteristic curve (AUC-ROC), to provide a more comprehensive evaluation of classifier performance.
Comments on the Quality of English Language
Can be improved
Reviewer 2 Report
Comments and Suggestions for Authors
The article addressed automated detection methods of the EEG for epilepsy. Thus, the deep learning approach for seizure detection and three EEG databases are studied. SHAP and LIME were utilized and showed some valid accuracy. Average accuracy vales are interesting to show it is unrelated. The references are well-provided. Figure quality is really good. English grammar has no issues. There are minor comments to be addressed before complete accpetance.
1. In Ref. section, abbreviated journal names are preferrable.
2. Is there any future work after this manuscript in detail ? What are variables ?
3. In Line 132, Pubmed -> PubMed.
4. In Line 134, Open access -> Open-access.
5. In conclusion section, there is unnecessary space between the sentences.
Reviewer 3 Report
Comments and Suggestions for Authors
A state-of-the-art deep-learning model for seizure detection and three EEG databases were chosen in this paper. Then, the models were trained and evaluated under different conditions (i.e., three distinct levels of overlap among the chosen EEG data windows.). Once the classifiers with the best performance were selected, Shapley Additive Explanations (SHAP) and Local Interpretable Model-agnostic Explanations (LIME) were employed to estimate the importance value of each EEG channel. Then, Spearman’s rank correlation coefficient between EEG features of epileptic signals and the importance values was computed. Results showed that the database and training conditions may affect classifiers’ performance. The most significant accuracy was 0.84, 0.73, and 0.64 for CHB-MIT, Siena, and TUSZ EEG datasets, respectively. In addition, most EEG features displayed a negligible and low correlation with the importance values. Finally, it is concluded that a correlation between the EEG features and importance values (generated by SHAP and LIME) may be absent, even for the high-performant models.
This is an interesting study, but there are still some issues that need to be addressed before publication.
1 To verify the generalization of the model, authors should use more baseline datasets.
2 Authors should compare more recently published methods as baseline methods.
3 Authors should discuss more EEG related work in the introduction, such as [1-2].
[1] Representation based on ordinal patterns for seizure detection in EEG signals. Computers in Biology and Medicine 2020
[2] MetaEmotionNet: Spatial-Spectral-Temporal based Attention 3D Dense Network with Meta-learning for EEG Emotion Recognition. TIM 2023
Comments on the Quality of English LanguageModerate editing of English language required
Round 2
Reviewer 1 Report
Comments and Suggestions for Authors
Suggestions are incorporated
Comments on the Quality of English LanguageNA
Reviewer 3 Report
Comments and Suggestions for Authors
The authors have solved my problem.
Comments on the Quality of English LanguageMinor editing of English language required